# Efficient Degradation of 2-Mercaptobenzothiazole and Other Emerging Pollutants by Recombinant Bacterial Dye-Decolorizing Peroxidases

**DOI:** 10.3390/biom11050656

**Published:** 2021-04-29

**Authors:** Aya Alsadik, Khawlah Athamneh, Ahmed F. Yousef, Iltaf Shah, Syed Salman Ashraf

**Affiliations:** 1Department of Chemistry, College of Arts and Sciences, Khalifa University of Science and Technology, Abu Dhabi P.O. Box 127788, United Arab Emirates; 100049511@ku.ac.ae (A.A.); khawlah.athamneh@ku.ac.ae (K.A.); ahmed.yousef@ku.ac.ae (A.F.Y.); 2Center for Membranes and Advanced Water Technology (CMAT), Khalifa University of Science and Technology, Abu Dhabi P.O. Box 127788, United Arab Emirates; 3Department of Chemistry, College of Sciences, UAE University, Al Ain P.O. Box 15551, United Arab Emirates; altafshah@uaeu.ac.ae; 4Center for Biotechnology (BTC), Khalifa University of Science and Technology, Abu Dhabi P.O. Box 127788, United Arab Emirates

**Keywords:** pollutants, wastewater, bacterial peroxidases, DyPs, LCMSMS

## Abstract

In recent years, concerns are being raised about the potential harmful effects of emerging pollutants (EPs) on human and aquatic lives. Extensive research is being conducted on developing efficient remediation strategies to target this new class of toxic pollutants. Studies focused on biological (enzyme-based) methods have shown potential as greener and possibly more economical alternatives to other treatment approaches, such as chemical methods. The current study focused on the use of recombinantly produced novel bacterial peroxidases, namely dye-decolorizing peroxidases (DyPs), to study their effectiveness in degrading a number of diverse EPs. In this context, a sensitive bioanalytical Liquid chromatography—tandem mass spectrometry (LCMSMS)-based method was developed to simultaneously detect a mixture of 31 EPs and to examine their degradability by a panel of seven different recombinant bacterial DyPs (rDyPs). We show that up to 8 of the 31 tested EPs could be degraded by at least one of the DyPs tested. The results also indicated that not all rDyPs behaved similarly in their abilities to degrade EPs, as some rDyPs (such as *Svi*DyP and *Cbo*DyP) showed a promising potential to degrade EPs while others (such as *Sc*DyP) were almost ineffective. Additionally, the role of redox mediators for effective emerging pollutant degradation by rDyPs was also examined, which showed dramatic improvement in the DyP-mediated degradation of five different EPs. Detailed analysis of 2-mercaptobenzothiazole degradation by *Svi*DyP showed that six distinct breakdown products were generated. The present study showed for the first time that recombinant bacterial DyPs can be used for wastewater remediation by degrading a range of different EPs.

## 1. Introduction

The past few years have witnessed the emergence of relatively new classes of anthropogenic pollutants entering the aqueous environment, particularly our drinking water supplies [1]. They have been grouped as emerging pollutants (EPs)—potentially toxic chemicals including antibiotics, steroids, hormones, personal care products (PCPs), pesticides, surfactants, and many others [2]. Current findings have pointed out their significant impact on human health ranging from minor effects to severe health conditions, including growth retardation and genotoxic effects [3,4].

Substantial efforts are being made to alleviate the impact of EPs on the environment by developing various wastewater treatment technologies [5,6,7,8]. Physical and chemical treatment techniques have been most commonly explored for wastewater treatment studies [9]. For example, removing EPs from textile effluents and real wastewaters can be easily carried out using physical processes including adsorption on powdered activated carbon [7,10]. Chemical treatments have benefited from the high reactivity of oxidizing species that can react with pollutants and eventually degrade them into less toxic compounds [11]. Advanced oxidation processes (AOPs) have become one of the most successfully established chemical treatment systems due to their efficiency and ease of use [12]. A combination of different AOPs, such as Fenton with ultrasound, has been shown to be even more efficient in degrading the emerging pollutant sulfadiazine [13].

Recent water remediation studies have also placed a great emphasis on the role of green chemistry in the degradation of EPs by exploring the potential of enzymes in remediating contaminated water bodies [14,15]. Owing to their abilities to degrade a range of EPs, including pharmaceuticals and hormones, peroxidases have increasingly gained the interest of the scientific community over the past few years [14,16]. Peroxidases are known biocatalytic agents that share a common reaction cycle with a unique functionality represented by the iron (Fe^3+^) of the heme center of the resting state of the enzyme. This metal ion can react with hydrogen peroxide (H_2_O_2_), which acts as an oxidizing agent to form an oxo-Fe^4+^^•^ cation radical of the enzyme, generating so called Compound I. This radical species can attack another species that exists in the reaction medium. Such species could be an organic substrate which generates an organic radical (R^•^), resulting in a less oxidized (more reduced oxo-Fe^4+^ form) form of the enzyme. This compound, which is named Compound II, can itself react with another organic substrate molecule to return to the resting form and create another organic radical (R^•^), as shown below:Peroxidase (resting state: Fe^3+^) + H_2_O_2_ → Peroxidase (Compound I: oxo-Fe^4+•^) + H_2_O
Peroxidase (Compound I: oxo-Fe^4+•^) + RH → Peroxidase (Compound II: oxo-Fe^4+^) + R^•^
Peroxidase (Compound II: oxo-Fe^4+^) + RH → Peroxidase (resting state: Fe^3+^) + R^•^ + H_2_O

The organic radicals (R^•^) produced in the reactions mentioned above can spontaneously react/breakdown to form smaller organic compounds [17].

A significant number of articles have focused on soybean and horseradish peroxidases (SBP and HRP, respectively), which have been among the top candidates that are being explored in this area [18,19]. A large number of reports have demonstrated promising potential for peroxidases for degrading a number of emerging pollutants [20,21,22]. Redox mediators, which are small, diffusible, redox-active organic compounds that act as “go-between agents” in peroxidase-catalyzed reactions, have also been shown to improve the catalytic potential of peroxidases for degrading some EPs [18,23]. Due to their small size and redox potential, these redox mediators preferentially react with the peroxidases and are converted into high reactive radicals, which in turn can react and degrade different organic compounds or recalcitrant EPs. Additionally, recent studies on the comparative efficiencies of different peroxidases for degrading the same EPs have shown that not all peroxidases are equally effective remediation agents and offer a better and more comprehensive approach to potentially using these peroxidases for degrading EPs [24].

A systematic exploration of the ability of dye-decolorizing peroxidases (DyPs) to degrade EPs has yet to be reported. DyPs are a relatively new family of heme peroxidases that are found in fungi, archaea and bacteria [25,26]. They have been reported to exhibit a wide substrate specificity, which has allowed for their use in the degradation of a wide range of aromatic and nonaromatic substrates [25,27,28]. Bacterial DyPs are increasingly being discovered and are the most abundant members of the putative DyP family (PeroxiBase; http://peroxibase.toulouse.inra.fr/ accessed on 9 March 2021). The continuous advancement in recombinant DNA technology has improved the potential for the successful and cheap production of active recombinant DyPs in *Escherichia coli* systems [29,30,31]. It is interesting to note that, unlike their eukaryotic counterparts, such as SBP and HRP, which are very difficult to produce in active forms using bacterial expression systems, recombinant bacterial and fungal DyPs appear to be reasonably well expressed in *E. coli*. The aim of this study was to explore the bioremediation potential of seven different commercially available recombinant bacterial DyP peroxidases. Due to the fact that they are a recently discovered family of peroxidases, very little is known about their substrate specificity and potential for remediating water contaminated with various types of EPs [32]. In this work, we present for the first time data showing the potential of these commercial recombinant bacterial DyPs (*YfeX*, *Tfu*DyP, *Pf*DyP B2, *Sc*DyP, *Tc*DyP, *Svi*DyP, and *Cbo*DyP) to simultaneously degrade a panel of 31 different EPs. This panel of EPs was chosen to represent a diverse range of structures and categories of organic pollutants (Table 1) that are commonly being detected in various water bodies [33,34,35].

## 2. Materials and Methods

### 2.1. Reagents

All 31 EPs listed in Table 1, as well as High-Performance Liquid Chromatography (HPLC) grade glacial acetic acid, 2,2’-azino-bis(3-ethylbenzothiazoline-6-sulfonic acid) (ABTS), and 1-hydroxybenzotraizole (HOBT), were purchased from Sigma-Aldrich (St. Louis, MO, USA). Liquid chromatography coupled with mass spectrometry (LCMS) grade solvents including methanol, acetonitrile, formic acid, absolute ethanol, and water were purchased from Millipore (Burlington, MA, USA). The seven DyPs: *YfeX (Escherichia coli* O157), *Tfu*DyP (*Thermobifida fusca)*, *Pf*DyP B2 (*Pseudomonas Fluorescens* Pf0-1), *Tc*DyP (*Thermomondpora curvata*), *Sc*DyP (*Streptomyces coelicolor*), *Svi*DyP *(Saccharomonospora viridis* DSM 43017), and *Cbo*DyP (*Cellulomonas bogoriensis*), were purchased from Gecco Biotech (Groningen, The Netherlands) as lyophilized powders that were dissolved in water.

### 2.2. Sequence Alignment of Seven Recombinant Bacterial DyPs (rDyPs)

The amino acid sequence of the seven DyPs was obtained from Gecco Biotech (Groningen, The Netherlands). The alignment was performed using Vector NTI Express software by Thermo Fisher Scientific (version number: 1.6.1) (Waltham, MA, United States). The phylogenetic tree for these seven rDyPs was generated using the Basic Local Alignment Search Tool (BLAST) developed by the National Center for Biotechnology Information (NCBI) (available online: https://www.ncbi.nlm.nih.gov/ accessed on 09 March 2021). The amino acid alignment of *Svi*DyP and *Cbo*DyP and the remaining rDyPs was also carried out using BLAST search.

### 2.3. Determination of Optimal pH for the Seven rDyPs

Universal buffer (0.2 M disodium phosphate (K_2_HPO_4_) and 0.1 M citric acid) was used to prepare buffers at pH values of 3, 4, 5, or 6. The pH activity profile of the different DyPs was assayed by initially supplementing these different buffers with 75 nM DyP, and 1 mM ABTS substrate. Upon the addition of H_2_O_2_ (0.25 mM), the kinetics of the oxidization of ABTS was monitored over time by measuring absorbance at 420 nm using a Carry 60 spectrophotometer (Agilent Technologies, Santa Clara, CA, USA) with 1 cm path length in a 2 mL cuvette. The linear portion of the absorbance (time-course) curves was used to calculate the slope, which corresponds to the rate (absorbance/min) of the enzyme at each tested pH. Each experiment was performed in triplicate.

### 2.4. Degradation of Emerging Pollutants by Recombinant DyPs

DyP-based degradation assays were conducted as previously described [24]. Briefly, two sets of reactions were set-up for each degradation study, one was prepared without the redox mediator, HOBT, while the other one contained HOBT. A mixture of 31 EP mixture (each at 4 ppm), specific DyP, and universal buffer were all added together. The degradation reaction was initiated by the addition of 0.25 mM H_2_O_2_. The enzyme (rDyPs) concentrations used were as follows: [*YfeX*] = 75 nM, [*Tfu*DyP] = 87 nM, [*Tc*DyP] = 750 nM, [*Sc*DyP] = 2250 nM, [*Svi*DyP] = 2813 nM, and [*Cbo*DyP] = 4328 nM. The concentrations of the DyPs were normalized based on ABTS-oxidation assays, and equal amounts of “peroxidase activity” were used for each DyP degradation assay. Where needed, 0.1 mM HOBT was included in the reaction mixture. Samples were filtered using 0.45 μm filters, prior to injecting them into the LCMSMS system.

### 2.5. LCMSMS Based EPs Degradation Assay

LCMSMS was used to quantify the 31 EPs before and after the DyP-mediated degradation. A sensitive and selective method based on a previously published study [36] was used, where the LCMSMS (Agilent 6420 Triple Quadrupole Mass Spectrometer (Agilent technologies, Santa Clara, CA, USA) was used in the multiple reaction monitoring (MRM) mode to simultaneously detect and quantify 31 EPs in a mixture. The MRM mode is capable of simultaneously detecting “precursor-to-product” ion transitions of an individual compound after being fragmented into specific fragments. The schematic of the method development is shown in Appendix A, where the MRM parameters are identified for Biochanin A (one of the 31 EPs analyzed in the current study). The first step in MRM method development is the confirmation of the “parent mass/charge” ratio by using the LCMSMS in the “full scan mode”. Once the parent ion (285 *m*/*z* for Biochanin A) is confirmed, it is then fragmented by increasing the collision energy in the Q2 of the LCMSMS—step 2. The last step uses the optimum collision energy from the 2nd step at which a strong fragmented product (152 *m*/*z* for a Biochanin A fragment) is detected to monitor the “precursor-to-product” (285 → 152 *m*/*z*) transition. The MRM parameters of all 31 EPs are shown in Table 1.

The separation of all the 31 EPs from the mixture was carried out through a dual solvent-gradient pump system, using a ZORBAX Eclipse plus C18 column maintained at 35 °C, with a particle size of 1.8 μm and column dimensions of 1.2 mm x 50 mm. The two solvents used for the sample elution were solvent A, which refers to 0.1% formic acid in water, and solvent B, represented by 100% acetonitrile at a flow rate of 0.4 mL/min over a 30-min run. The run started with the gradient of 95% A and 5% B for 5 min. The percentage of B then increased from 5% to 90% over 20 min. In the next 10 s, B% dropped from 90% to 5% over the next 5 min. A dual-polarity electrospray ionization (ESI) source was used to ionize the eluted compounds in both positive and negative polarity modes. The mass spectrometry operating parameters for capillary voltage, nebulizer pressure, temperature of the gas and gas flow rate were set to 4000 V, 45 psi, 8 L/min, and 325 °C, respectively, as previously reported [36].

When MRM chromatograms were generated, the extracted peaks representing the MRM transitions of an individual EP were used to quantify the pollutants remaining after the enzymatic treatment. This was achieved by measuring the areas under the curves (AUCs) of individually extracted EPs from the MRM chromatograms. The AUCs of the EPs obtained with and without (controls) enzymatic treatment were used for measuring their degradation efficiencies and percentage remaining according to the following equation:$$\%\;\mathrm{Percentage}\;\mathrm{Remaining}:\;\frac{\mathrm{AUC}\;}{\mathrm{AUC}\;\left(\mathrm{control}\right)}\;\times \;100$$

AUC: treated sample containing DyP, H_2_O_2_, buffer and EP (± HOBT) and AUC (control): DyP, buffer and EP (± HOBT).

### 2.6. LCMSMS Analysis of Products of MBT Degrdation

Identification of possible intermediates formed during *Svi*DyP-mediated degradation of pollutants was conducted using LCMSMS in full scan mode as previously described [24]. MBT degradation product identification experiments were carried out based on the conditions established for *Svi*DyP as described earlier, except MBT was used at 100 ppm. The reaction was initiated with a sequential additon of 0.25 mM H_2_O_2_ three times at 20-min intervals to the reaction mixture (total reaction time was 60 min). The samples were then analyzed using the full-scan mode on the LCMSMS over a mass range of 50-1000 **m*/*z**. The Agilent MassHunter Qualitative Analysis tool was used to extract and analyze the data for all possible full scan and MRM runs.

## 3. Results and Discussion

### 3.1. Sequence Alignments of the Recombinant Bacterial DyPs

An amino acid sequence alignment of the seven DyPs that were used in this study is presented in Figure 1. As can be seen from the figure, some residues are highly conserved among the seven enzymes forming a GXX(D/E)G motif, which is commonly found among DyP members [28]. Six enzymes, namely *YfeX*, *Tfu*DyP, *Pf*DyP B2, *Sc*DyP, *Tc*DyP, and *Svi*DyP, were found to have an aspartate residue, two amino acids down from the glycine, whereas *Cbo*DyP was the only enzyme that had a glutamate at this position (Figure 1A). Nevertheless, the phylogenetic tree constructed from the amino acid sequences reveals that *Cbo*DyP shares a close common ancestor with *Svi*DyP when compared to the other enzymes, which suggests that these two enzymes could share similar functional properties (Figure 1A,B). In fact, as shown in panels C and D of Figure 1, *Cbo*DyP and *Svi*DyP share a high level of similarity (63% amino acid identity). By comparison, other DyPs had significantly lower similarity scores when compared to *Svi*DyP. The importance of the GXX(D/E)G motif found in DyPs in their peroxidase activity has already been established by a number of reports that described their relation to the enzymatic function of several DyPs [25,32,37].

### 3.2. Determination of Optimal pH for the Recombinant Bacterial DyPs

One of the important parameters that has a direct effect on the ionizable groups in the enzyme’s active site, their interaction with substrates, and catalytic activity is the pH of the enzymatic reaction. A crucial step to achieve optimum catalytic performance for an enzyme is to determine the optimum pH using a suitable buffer [38]. The pH optima for the seven bacterial rDyPs were determined through pH-dependent activity assays that were performed using ABTS as a model peroxidase substrate (Figure 2). Among the tested DyPs, *YfeX*, *Tfu*DyP, *Tc*DyP, *Sc*DyP and *Svi*DyP were most active at pH 4, while the best activities for *Pf*DyP B2 and *Cbo*DyP were observed at pH 3 and pH 5, respectively. This might be due to the presence of an aspartate residue in the GXXDG motif of the active site of the first six rDyPs, as previously described [32]. The observed optimum activity at pH 5 for *Cbo*DyP was also reported in a previous study that tested the enzyme’s activity on Reactive Blue 19 as a substrate [39]. Although DyP-mediated activity has been found to be substrate dependent [25], it has been postulated that the presence of glutamate in the active site plays a role in tuning the optimum pH for the enzyme [39]. Thus, it was expected that the highest activity of this enzyme would be observed at a slightly higher pH than that of other DyPs tested [39]. The lower optimum pH for *Pf*DyP (pH = 3) as compared to the pH 4 optimum for other active site aspartate-containing DyPs could be due to the microenvironment of the active site of *Pf*DyP, which could cause the aspartate carboxylate to have a lower pKa. It will be interesting to confirm this hypothesis with additional future studies.

### 3.3. DyPs-Mediated Degradation of Emerging Pollutants

LCMSMS has long been used to quantify organic compounds present in different matrices [40]. We have recently published on the potential use of a bioanalytical approach that can be used to simultaneously test the peroxidase-mediated degradation of a large number of organic pollutants in a mixture. The study has described the use of LCMSMS to develop an MRM method that can simultaneously detect 21 different EPs in solution [36]. A similar MRM approach was adopted in this study, which was based on using LCMSMS to develop an MRM method for 31 EPs belonging to different chemical classes. The names of these EPs, their categories and MRM parameters set for the developed method are presented in Table 1. The MRM chromatogram generated for the mixture of 31 EPs along with individual chromatograms extracted for some emerging compounds in the solution and their chemical structures are presented in Figure 3. Each compound was resolved based on detecting a particular MRM transition (precursor → product ion) which is specific for that compound (Appendix A). As described in the methods section, this LCMSMS-based approach was used to test the ability of the chosen seven DyPs to degrade a panel of 31 different emerging pollutants.

Table 2 represents the degradation profiles of all 31 EP compounds after treatment with the seven rDyPs. Notably, eight EPs were found to be efficiently degraded (25–75%) by these enzymes, with one or more emerging pollutants being degraded at least by one peroxidase (shaded area in Table 2). Additionally, other emerging pollutants showed almost 25% degradation, as clearly seen in the lightly shaded area in Table 2. As mentioned earlier, the EP classes were selected based on their reported detection in various water bodies as well as spanning a wide molecular weight range (71 g/mol to 836 g/mol). It is interesting to note that most of the degraded EPs appeared to be lower molecular weight compounds (around 300 g/mol), with lincomycin (407 g/mol) showing minor degradation and the highest molecular weight compound (roxithromycin, 836 g/mol) showing no degradation at all. Perhaps there is an upper size/mass limit to organic pollutants that can be degraded by these peroxidases. Studies like the present one that examine the peroxidase-mediated degradability of a large number of organic compounds belonging to diverse classes and structures are critical in elucidating potential “common structural elements” (if any) that may influence their bioremediation. In addition to structures and sizes, it appears that the redox potential of compounds may also play an important role in the peroxidase-mediated degradation of organic pollutants [17].

An interesting observation was that some of the rDyPs were more active on a given pollutant than others, for example, the degradation of 2-Mercaptobenzothiazole (MBT), a vulcanizing agent and water pollutant, varied dramatically depending on the DyP being used (Figure 4A,B; Table 2). MBT was completely degraded by *Svi*DyP and *Cbo*DyP, as evident by the disappearance of the peak after treatment, while treatment with *Pf*DyP B2 and *Tfu*DyP did not result in a significant degradation of the pollutant (Figure 4A,B). Similarly, such differential degradation of other emerging pollutants was observed for the DyPs tested. For example, Figure 5 shows extracted chromatograms of four emerging pollutants treated with *Cbo*DyP, *Svi*DyP and *Tfu*DyP. As shown in Figure 5A, complete degradation of MBT was achieved with *Cbo*DyP, whereas *Svi*DyP was most active on caffeic acid (causing around 70% degradation) (Figure 5B). Emerging pollutant Gemfibrozil, a common lipid regulating drug, was degraded most efficiently by *Tc*DyP (~70%), but *Tfu*DyP could only degrade approximately 25% of it (Figure 5C). Similarly, paracetamol seemed to be quite recalcitrant to degradation, and showed only slight degradation by *Svi*DyP (~15%), as shown in Figure 5D. Another observation from Table 2 is that the two DyPs, namely *Svi*DyP and *Cbo*DyP, seemed to have the most degrading potential among the seven tested DyPs. Interestingly, *Svi*DyP was previously described as having the potential to decolorize several triarylmethane dyes, anthraquinone and azo dyes [41]. It has been suggested that *Svi*DyP can exhibit substrate promiscuity, which can allow it to catalyze a wide variety of potentially unknown xenobiotics [41]. Indeed, other emerging pollutants such as Fluometuron, Venlafaxine, 3-Methyl-2-benzothiazolinone, and (4-Chloro-2-methylphenoxy) acetic acid were found to be degraded upon *Svi*DyP-mediated treatment, as seen in Table 2. Likewise, *Cbo*DyP was found to have the tendency to degrade these emerging pollutants as well. Such an interesting finding might be attributed to the similarity in their evolution profile and homology in their amino acid sequences, as shown in Figure 1, where both enzymes showed approximately 60% homology. A recent study has solved the crystal structure of *Cbo*DyP; however, the crystal structure of *Svi*DyP has not been released yet [39]. Further structural analysis may shed light on the possible reason behind their similar catalytic preferences.

### 3.4. The Role of Redox Mediating Species

It is well known that some peroxidase-mediated degradation of organic compounds can be dramatically enhanced by the addition of redox mediators [15,16]. In the current study, the commonly used redox mediator, HOBT, was used to assess its effect on the enzymatic degradation of EPs mediated by the selected rDyPs. Since the initial degradation of Prometryn by *Svi*DyP (in the absence of HOBT) showed only 10% degradation, we chose to test the degradation of this pollutant by *Svi*DyP in the presence of HOBT. As can be seen in Figure 6, the addition of HOBT caused about 40% of the Promteryn to be degraded (as compared to only 10% in the absence of HOBT). This HOBT-mediated enhancement of the enzyme’s ability to degrade pollutants has also been observed with other eukaryotic peroxidases [18,21]; for example, we have previously shown that adding HOBT significantly enhanced the degradation of sulfamethoxazole by SBP [18]. Additionally, when the HRP catalytic system was coupled with the redox mediator, ABTS, its capability in transforming tetracyclines in wastewater effluent was also enhanced [21]. The enhanced degradation of organic pollutants by redox mediators is attributed to the formation of redox mediator radical species that play a role in accelerating the degradation reaction. Our additional experiments with rDyPs showed that the degradation of thirteen emerging pollutants could be enhanced by the HOBT-mediated rDyP system (Table 3 and Appendix A). It is worth highlighting the interesting obervation that roxithromycin, the highest molecular weight (836 g/mol) EP tested in our study, which was completely recalcitrant to all the rDyPs tested, showed around 50% degradation with *Pf*DyP in the presence of HOBT (Table 3). However, as seen in our current study, the addition of a redox mediating species is not universally beneficial [36] and some redox mediating species exert no influence on the catalytic efficiency of the enzymes [36,42,43]. For example, the degradation of all pollutants by *Tc*DyP was unaffected when HOBT was added to the system (Appendix A). Additionally, it has been reported that, in some cases, the addition of redox mediators to an enzymatic system might, in fact, decrease the catalytic efficiency of the degradation due to competition for the active site between the redox mediator and organic pollutants [36]. Indeed, in the current study, the remediation of a few EPs by rDyPs was inhibited in the presence of HOBT (Appendix A).

### 3.5. LCMSMS Analysis of MBT Intermediates Generated by SviDyP

It is important to study the chemical fate of EPs degraded by water treatment technologies as the intermediate products that result from degradation could also be harmful to the environment and human health. Although there are many reports of remediation treatments reducing the toxicity of the parental toxic pollutant, a number of reports have shown that degradation products of toxic pollutants can still be toxic [17]. In the present study, we compared the degradation intermediates formed when MBT was degraded by *Svi*DyP (six different species, Table 4) with some of the products formed when MBT is degraded by either treatment method (including chemical methods and peroxidases with and without redox mediators). As summarized in Table 5, all of the MBT degradation products we found have been previously reported in literature [24,44,45,46,47]. A summary of these intermediates is presented in Figure 7, illustrating their potential structures as well as possible chemical reactions that could lead to their formation based on literature findings. The structure of the intermediate with 123 *m*/*z* was not proposed. Perhaps future structural analysis might be helpful to draw on the structure of this transformation product. Interestingly, a recent study on the degradation products generated during UV-treatment of MBT found a large number of them (different products than what we detected in our study) still had significant aquatic toxicity and even potential hazard to human health [48]. Again, this underlies the importance of toxicity studies as part of future remediation treatment studies.

## 4. Conclusions

The current study investigated the potential of using seven novel recombinant bacterial DyPs (expressed in *E. coli*) as biocatalytic agents for the degradation of a number of emerging pollutants as a mean of the bioremediation of polluted water. It was found that rDyPs exhibited differential abilities in degrading these pollutants, with recombinant *Svi*DyP appearing to be the most potent enzyme, being able to degrade 12 of the 31 EPs tested. *Svi*DyP was further used to degrade 100 ppm MBT, an emerging pollutant of major concern due to its toxicity against microorganisms and potential carcinogenicity [49,50]. Potential degradation products of MBT were identified in order to gain insight on the possible chemical pathways involved in degradation. The results presented here pave the way for the use of cheaply produced recombinant bacterial and fungal peroxidases for wastewater remediation applications. Additionally, mutational and enzyme engineering approaches can also be applied to these bacterial peroxidases to evolve them into more potent bioremediation agents [39].

## Figures and Tables

**Figure 1 biomolecules-11-00656-f001:**
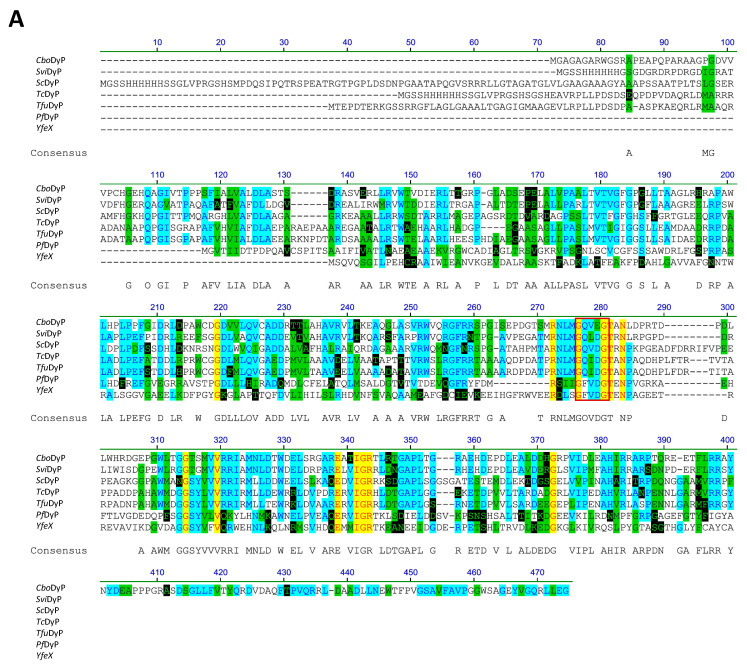
(**A**) Multiple sequence alignments of seven bacterial DyPs obtained using Vector NTI. *Cbo*DyP, *Svi*DyP, *Tc*DyP, *Tfu*DyP are A-type DyPs and *Sc*DyP, *Pf*DyP B2 and *Yfe*X are B-type DyPs. Residues in the alignment are colored according to the following scheme: Black on white = non-similar to other residues at that position. Blue on cyan = residues that are conserved in the majority of the DyPs at that position. Black on green = similar amino acids. Red on yellow = identical residues at that position. Green on black = residues that are weakly similar. Shown in the alignment as well is the conserved DyP motif “GXX(D/E)G” at position 276–280. (**B**) Phylogenetic tree of the seven rDyPs. (**C**) Sequence similarity analysis based on multiple sequence alignment for 7 DyPs, with 6 different DyPs compared with *Svi*DyP. Percent identity represents the percentage of amino acids that match up in the alignment and “Query coverage” shows the percentage sequence length that was included in the alignment analysis. (**D**) Sequence alignment of *Svi*DyP and *Cbo*DyP showing the similar amino acids between the two enzymes.

**Figure 2 biomolecules-11-00656-f002:**
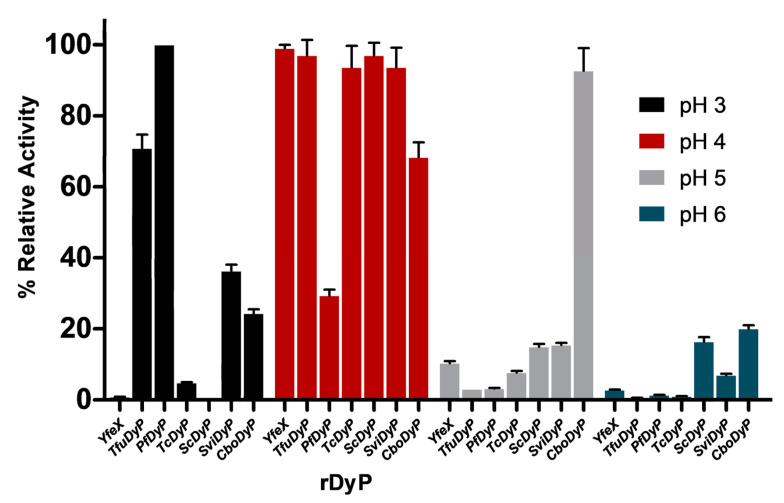
Activity profiles of the seven rDyPs obtained at four pH values using ABTS oxidation assay. [rDyP] = 75 nM, [ABTS] = 1 mM, [H_2_O_2_] = 0.25 mM, universal buffer.

**Figure 3 biomolecules-11-00656-f003:**
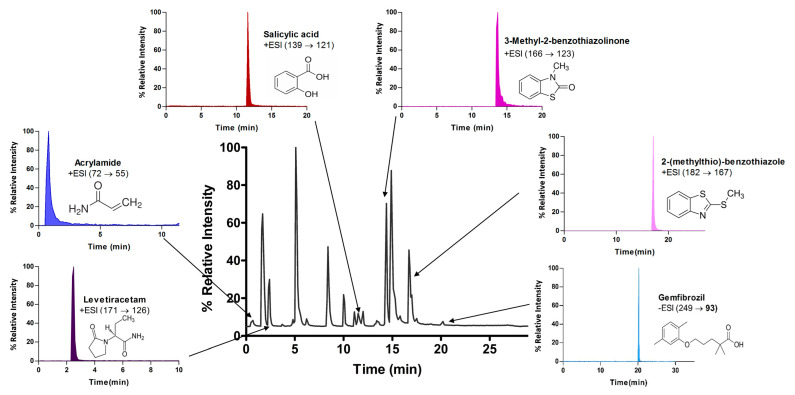
Typical combined LCMSMS MRM chromatogram of the mixture of 31 emerging pollutants at 4 ppm of each of them. The representative extracted MRM chromatograms of six chosen emerging pollutants (EPs) are also shown.

**Figure 4 biomolecules-11-00656-f004:**
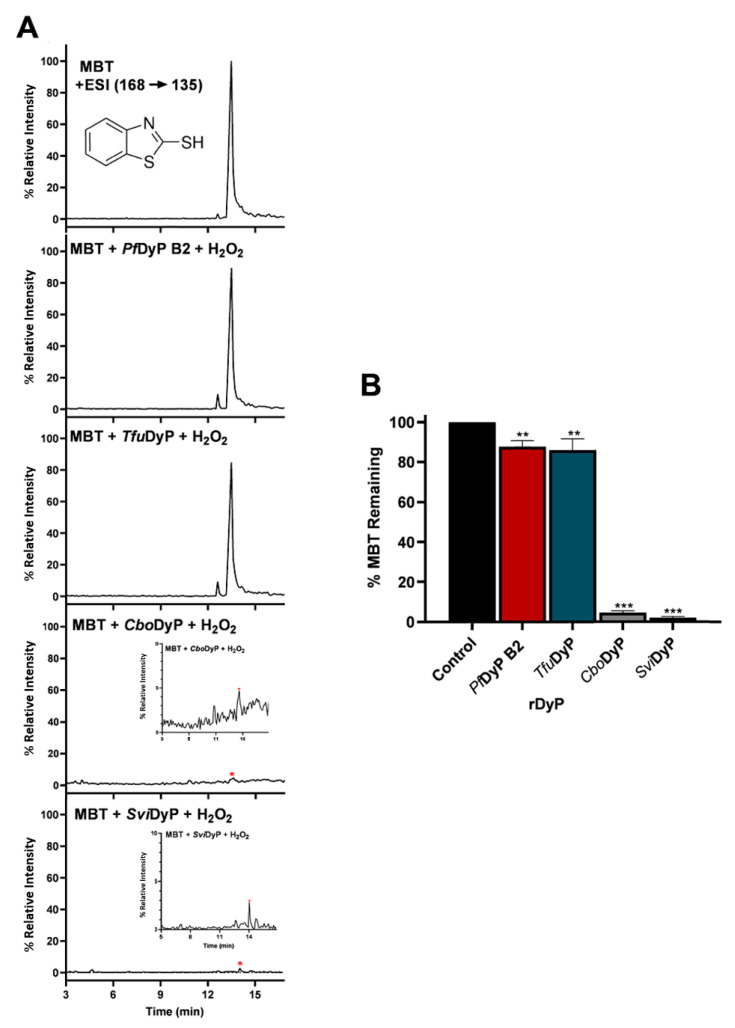
(**A**) MRM chromatograms of MBT untreated and treated samples with *Pf*DyP B2, *Tfu*DyP, *Cbo*DyP and *Svi*DyP. (**B**) Percentage of MBT remaining upon degradation by *Pf*DyP B2, *Tfu*DyP, *Cbo*DyP and *Svi*DyP. Reaction conditions: [EP] = 4 ppm, [*Pf*DyP B2] = 0.255 µM, [*Tfu*DyP] = 0.087 µM, [*Cbo*DyP] = 4.33 µM, [*Svi*DyP] = 2.8 µM, [H_2_O_2_] = 0.25 mM, pH values for *Pf*DyP B2, *Tfu*DyP, *Svi*DyP, and *Cbo*DyP are 3, 4, 4, and 5, respectively. Insets in the *Cbo*DyP and *Svi*DyP panels show zoomed in chromatograms with the red asterisk showing the residual MBT. ** *p* < 0.01 *** *p* < 0.001 compared to the “Control”.

**Figure 5 biomolecules-11-00656-f005:**
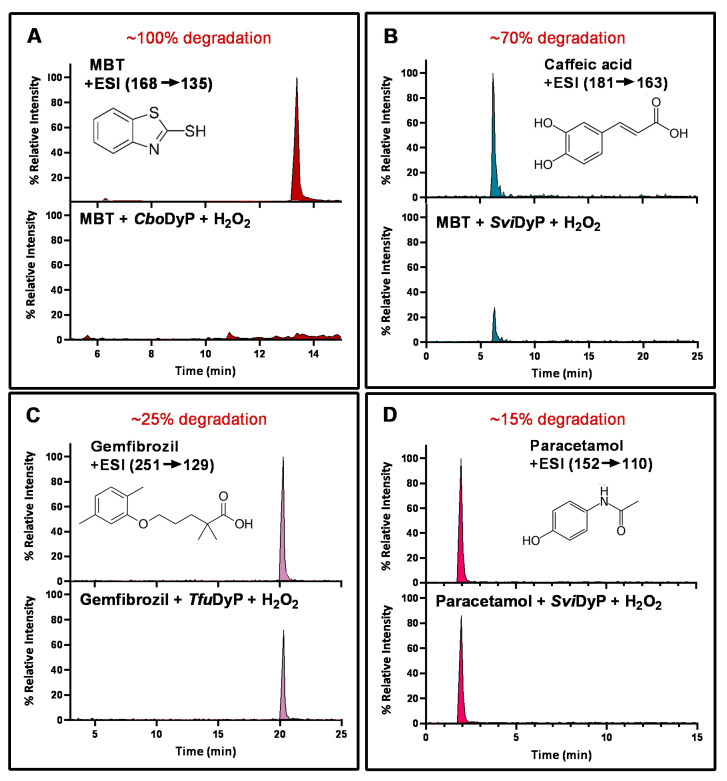
Individual MRM chromatograms of four emerging pollutants; their chemical structures and degradation percentages upon DyP treatment. (**A**) *Cbo*DyP-untreated and treated MBT, (**B**) *Svi*DyP-untreated and treated Caffeic acid, (**C**) *Tfu*DyP-untreated and treated Gemfibrozil and (**D**) *Svi*DyP-untreated and treated Paracetamol. Reaction conditions: [EP] = 4 ppm, [*Svi*DyP] = 2.8 µM, [*Cbo*DyP] = 4.33 µM, [*Tfu*DyP] = 0.087 µM, [H_2_O_2_] = 0.25 mM, their optimum pH = 4, 5 and 4, respectively.

**Figure 6 biomolecules-11-00656-f006:**
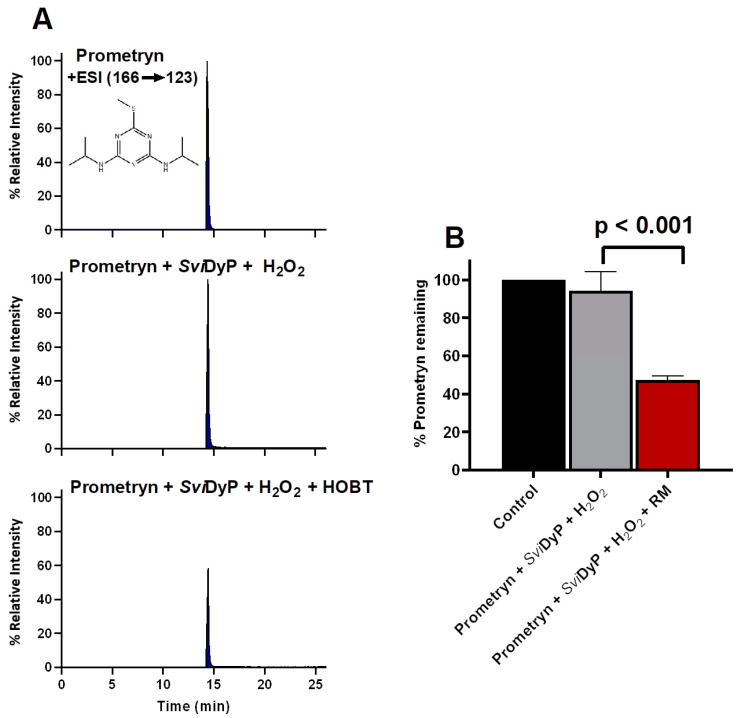
(**A**) MRM chromatograms extracted for Prometryn treated with *Svi*DyP with and without HOBT. (**B**) shows the percentage of Prometryn remaining after both treatments. Reaction conditions: [EP] = 4 ppm, [*Svi*DyP] = 2.8 µM, [H_2_O_2_] = 0.25 mM, pH = 4.

**Figure 7 biomolecules-11-00656-f007:**
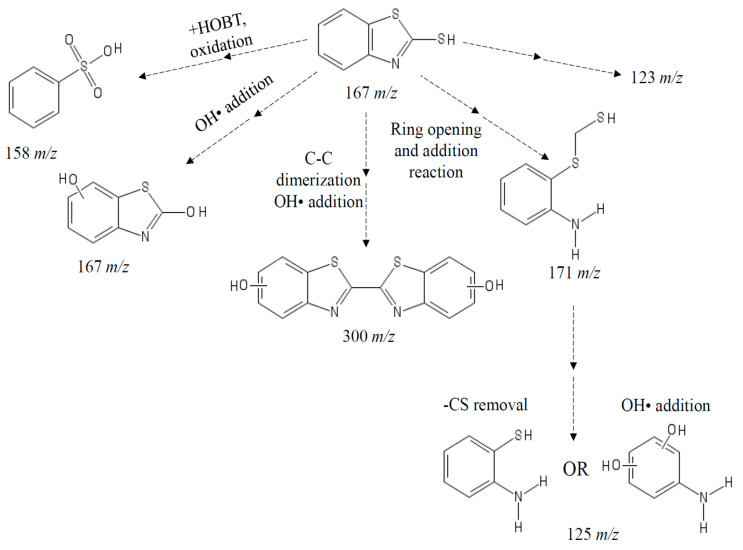
The structures of proposed MBT intermediates generated after *Svi*DyP-mediated treatment and potential pathways for the formation of the intermediates.

**Table 1 biomolecules-11-00656-t001:** Summary of the chemical structures as well as the LCMSMS parameters for the EPs used in the current study.

#	Category	Emerging Pollutants (EPs)	Structure	Retention Time(min)	MRM Transition (*m*/*z*)	Fragmentor Voltage (V)	Collision Energy (V)	Polarity
1	Antibiotic	Sulfamethoxazole	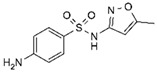	11.2	254 → 156	135	20	Positive
2	Antibiotic	Trimethoprim	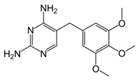	8.4	291 → 230	135	20	Positive
3	Antibiotic	Norfloxacin	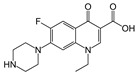	10.1	320 → 302	135	20	Positive
4	Antibiotic	Chloram-phenicol	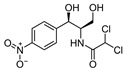	12.1	321 → 152	0	20	Negative
5	Antibiotic	3-Methyl-2(3H)-benzothiazolone	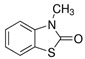	13.7	202 →175	135	30	Positive
6	Antibiotic	Penicillin GK	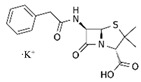	13.7	335 → 160	0	10	Positive
7	Antibiotic	Lincomycin-HCl	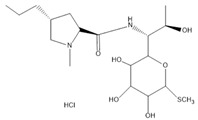	4.8	407 → 359	135	20	Positive
8	Antibiotic	Roxithro-mycin	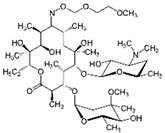	15.0	837 → 680	135	20	Positive
9	Anti-oxidant	Caffeic acid	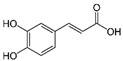	6.4	181 → 163	135	20	Positive
10	Anti-seizure drug	Levetiracetam	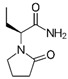	2.5	171 → 126	0	10	Positive
11	Anti-seizure drug	Phenytoin	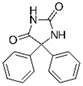	14.4	253 → 182	135	10	Positive
12	Anti-depressant	Venlafaxine-HCl	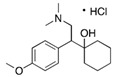	12.0	278 → 260	135	10	Positive
13	Beta-blocker	Atenolol	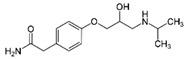	1.6	171 → 126	0	10	Positive
14	Diuretic drug	Hydrochloro-thiazide	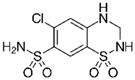	3.7	296 → 269	140	20	Negative
15	Diuretic drug	Furosemide	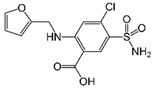	14.6	329 → 285	140	15	Negative
16	Flocculation agent	Acrylamide	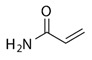	0.8	72 → 55	50	10	Positive
17	Fungicide (or its derivative)	Thiabendazole	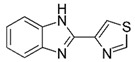	5.2	267 → 190	135	20	Positive
18	Herbicide	Prometryn	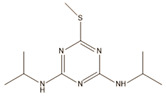	14.4	166 →123	135	30	Positive
19	Herbicide	2-methyl-4-chlorophenoxyacetic acid (MCPA)	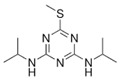	15.8	242 → 158	135	30	Positive
20	Herbicide	Fluometuron	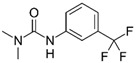	15.0	201 → 125	47	13	Positive
21	Histamine H₂ receptor antagonist	Cimetidine	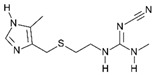	1.6	233 → 72	135	30	Positive
22	Insect repellent	N, N-Diethyl-meta-toluamide (DEET)	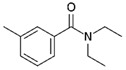	15.3	192 → 119	135	30	Positive
23	Keratolytic agent	Salicylic acid	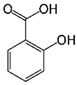	11.6	139 → 121	50	10	Positive
24	Lipid regulating agent	Gemfibrozil	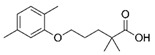	20.3	251 → 129	0	40	Positive
25	Non-steroidal anti-inflammatory drugs (NSAID)	Meloxicam	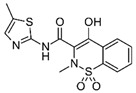	16.7	352 → 115	135	6	Positive
26	NSAID	Ibuprofen	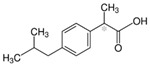	19.0	207 → 161	135	20	Positive
27	Phyto estrogen	Biochanin A	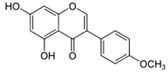	17.5	285 → 152	0	30	Positive
28	Pain killer	Paracetamol	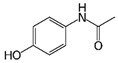	1.9	152 → 110	0	10	Positive
29	Rubber additive	2-(methylthio) benzothiazole (MTBT)	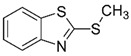	17.1	182 → 167	135	30	Positive
30	Stimulant	Caffeine	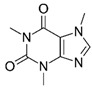	6.8	195 → 138	135	30	Positive
31	Vulcanization agent	2-Mercapto benzothiazole (MBT)	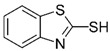	13.3	168 → 135	135	30	Positive

**Table 2 biomolecules-11-00656-t002:** Degradation profiles of 31 EPs upon treatment by seven rDyPs, as described under Materials and Mathods. EPs remaining after enzymatic treatment are indicated as follows: (++++) = 0–25% remaining of pollutant, (+++) = 25–50%, (++) 50–75% remaining, (+) = 75–90% remaining, (-) = < 90% remaining. Rows shaded in light red show most pronounced degraded EPs, those in light yellow represents EPs showing relatively low degradation, while EPs in unshaded rows showed no detectable degradation by any of the DyPs.

	DyPs	*YfeX*	*Tfu*DyP	*Pf*DyP B2	*Tc*DyP	*Sc*DyP	*Svi*DyP	*Cbo*DyP
31 EPs								
2-Mercaptobenzothiazole		-	+	+	-	-	++++	++++
Gemfibrozil		++	+	+	+++	-	-	-
Caffeic Acid		-	-	-	-	-	+++	++
Acrylamide		++	++	++	+	-	+	-
Biochanin A		++	+	+	-	+	-	-
3-Methyl-2-benzothiazolinone		-	-	-	-	-	+	++
(4-Chloro-2-methylphenoxy) acetic acid		-	-	-	-	-	+	++
Venlafaxine		-	-	-	-	-	++	-
Ibuprofen		+	+	+	-	-	+	-
Fluometuron		-	-	-	-	-	+	+
Cimetidine		-	+	+	-	+	-	-
Salicylic acid		-	-	-	-	-	+	+
Chloramphenicol		-	-	-	+	+	-	-
Lincomycin hydrochloride		-	-	-	-	-	+	-
DEET		-	-	-	-	-	-	+
Paracetamol		-	-	-	-	-	+	-
2-(Methylthio) benzothiazole		-	+	-	-	-	-	-
Sulfamethoxazole		-	-	-	-	-	+	-
Levetiracetam		-	-	-	-	-	-	-
Caffeine		-	-	-	-	-	-	-
Thiabendazole		-	-	-	-	-	-	-
Prometryn		-	-	-	-	-	-	-
Phenytoin		-	-	-	-	-	-	-
Atenolol		-	-	-	-	-	-	-
Trimethoprim		-	-	-	-	-	-	-
Hydrochlorothiazide		-	-	-	-	-	-	-
Furosemide		-	-	-	-	-	-	-
Penicillin		-	-	-	-	-	-	-
Meloxicam		-	-	-	-	-	-	-
Roxithromycin		-	-	-	-	-	-	-

**Table 3 biomolecules-11-00656-t003:** Selected emerging pollutants that showed an enhanced degradation after treatment by specific rDyPs in the presence of the redox mediator, HOBT.

rDyPs	Emerging Pollutants	% Remaining (No HOBT)	% Remaining (+ HOBT)
*YfeX*	Phenytoin	-	++
*Pf*DyP B2	Gemfibrozil	+	+++
	Roxithromycin	-	++
*Svi*DyP	Prometryn	-	+++
*Cbo*DyP	Penicillin	-	++

**Table 4 biomolecules-11-00656-t004:** Mass to charge ratios (*m*/*z*) of intermediates generated during *Svi*DyP-mediated treatment of MBT with and without redox mediator, HOBT.

Intermediates (*m*/*z*)	Without HOBT	With HOBT
123	√	√
125	√	√
158	Not detected	√
167	√	√
171	√	√
300	√	√

**Table 5 biomolecules-11-00656-t005:** Previously and currently reported intermediates of MBT generated during biological and chemical treatments.

Biological	Chemical	Process/Agent	Ion Mass (*m*/*z*)	References
√		Bacterial strain, ***Alcaligenes*** sp. ***CSMB1***	95, 106, **123**, 136, 150, 151, 165	[44]
√		SBP	120, 136, 182, 301, 332	[24]
		CPO		
√		*Svi*DyP	**123, 125, 158, 167 171, 300**	This study
	√	Photodegradation by Fe_3_O_4_-QDs@g-C_3_N_4_/ATP	82, 110, 114, **125**, 146, **171**	[45,46]
	√	Photodegradation by 9- Bi_2_WO_6_/In (OH)_3_ composite		
√	√	***Eurobacteria*** and Graphene-based anode and stainless-steel cathode	93, 94, 108, 109, 110, **125**, 126, 135, 151, **158**, **167**, 169, 174, 183, 187, 199, 215, 231, 268, 283, 284, **300**, 332, 364	[47]

## Data Availability

All relevant data are included in the paper or its Supplementary Information.

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
