# Peer review of "Efficient Degradation of 2-Mercaptobenzothiazole and Other Emerging Pollutants by Recombinant Bacterial Dye-Decolorizing Peroxidases"

_biomolecules, 2021, doi:10.3390/biom11050656_

Round 1
Reviewer 1 Report
Major comments:
The study under the title “Efficient degradation of 2-Mercaptobenothiazole and other emerging pollutants by recombinant bacterial dye-decolorizing peroxidases”. This paper brings an important contribution to the degradation and adsorption field. The manuscript is well compiled. The theme is interesting and presenting a timely effort from authors. However, the literature review needs to be updated with more recently published research. Promising data is presented in the paper, however, the manuscript is not equipped with any figure or visual representations. The use of a full form of abbreviations at first sight is required. This manuscript can be accepted for publication after some following careful revision.
Introduction:
The introduction needs to be more emphasized on the research work with the detailed explanation of the whole process considering past, present and future scope. The conventional methods to be explained well to indicate the relevance of the research work. It needs to be strengthened in terms of recent research and updated literature review in this area with possible research gaps. It is strongly recommended to add a recent literature survey about different types of wastewater treatment techniques, dye-decolorizing, nanomaterials, nanocomposites for efficient degradation with their applications in the various industrial sectors. Research gaps should be highlighted more clearly and future applications of this study should be added.
Specific comments:
1. The abstract only contains some parameters without any process conditions or key values from results, which is insufficient to delineate the whole pictures of contribution and possible application of this study. Therefore, it is suggested to add some background with few objectives and possible applications of this study and highlight the novelty of this work clearly.
2. Page 2: “A significant number of articles have focused on soybean and horseradish peroxidases (SBP and HRP, respectively), which have been among the top candidates that are being explored in this area”. How do these mentioned peroxidases help in degradation? Explain this term in the text.
3. Page 2: “Redox mediating species have also been used to improve the catalytic potential of some peroxidases for degrading some EPs”. How redox mediating species improve the potential of some peroxidases for degrading some EPs? Explain this term in the text. Also categorized the peroxidases and Eps regarding this context.
4. Page 2: “The continuous advancement of recombinant DNA technology has improved the potential for the successful and cheap production of active recombinant bacterial DyPs in Escherichia coli systems, unlike their eukaryotic counterparts, such as SBP and HRP, which are very difficult to produce in active forms using bacterial expression systems”. How these calculations have been performed explain thoroughly? Try to write a full form of abbreviation at the first sight.
5. Page 3: “After subjecting the EPs mixture to treatment with the seven rDyPs, samples were filtered using 0.45 μM 94 filters, prior to injecting them into the LCMS system”. Explain about the LCMS also write a full form of abbreviation at the first sight.
6. Page 4: “A 30-min elution run (in the MRM method) was developed where (i) 95% A and 5% B gradient was set for 5 min. (ii) a 5-90 % gradient of B was then adjusted for 20 min (from 5 min (95 % A) to 25 min (10 % A)) and another gradient of B % from 90 % 102 to 5% set for 10 seconds. (iii) at 25.10 min, 95 % A and 5 % B was set as a gradient for 4.90 min”. What is the MRM methods explain thoroughly?
7. Page 4: “The Mass spectrometry operating parameters for capillary voltage, nebulizer pressure, temperature of the gas and gas flow rate were set to 4000 V, 45 psi, 8 L/min, and 325 °C, respectively as previously reported”. Please site the relevant reference.
8. More recent research about types of wastewater pollution, treatment methods, various bi-composites and pollution reduction technologies is suggested to be added to make the background and discussion more strong: Journal of Water Process Engineering, 2020; 33:101101. Journal of Molecular Liquids, 2020; 313:113494. Journal of Environmental Chemical Engineering, 2020 Sep 19:104502. Journal of Environmental Management, 2020; 278:111302. International Journal of Biological Macromolecules, 2020; 161:1272-1285
9. Page 6: “An amino acid sequence alignment of the seven DyPs that were used in this study are presented in Figure 1”. Figure 1 is missing in this Manuscript.
10. The authors are advised to write the conclusions in a comprehensive way and should contain key values, suitability of the applied method, the major findings and contributions.
11. The manuscript does not contain any figure. Please add some figures or convert table data into figures.
12. The authors are advised to revise references, including the latest references. Please see some suggestions in the comments for the ‘introduction’ section.
Author Response
Response to reviewers
Re: Biomolecules-1158090
Reviewer 1:
The study under the title “Efficient degradation of 2-Mercaptobenothiazole and other emerging pollutants by recombinant bacterial dye-decolorizing peroxidases”. This paper brings an important contribution to the degradation and adsorption field. The manuscript is well compiled. The theme is interesting and presenting a timely effort from authors. However, the literature review needs to be updated with more recently published research. Promising data is presented in the paper, however, the manuscript is not equipped with any figure or visual representations. The use of a full form of abbreviations at first sight is required. This manuscript can be accepted for publication after some following careful revision.
Introduction:
The introduction needs to be more emphasized on the research work with the detailed explanation of the whole process considering past, present and future scope. The conventional methods to be explained well to indicate the relevance of the research work. It needs to be strengthened in terms of recent research and updated literature review in this area with possible research gaps. It is strongly recommended to add a recent literature survey about different types of wastewater treatment techniques, dye-decolorizing, nanomaterials, nanocomposites for efficient degradation with their applications in the various industrial sectors. Research gaps should be highlighted more clearly and future applications of this study should be added.
Specific comments:
1. The abstract only contains some parameters without any process conditions or key values from results, which is insufficient to delineate the whole pictures of contribution and possible application of this study. Therefore, it is suggested to add some background with few objectives and possible applications of this study and highlight the novelty of this work clearly.
Response: Thank you for highlighting this. As requested, the abstract has been revised to include specific details and results from the study.
Page 2: “A significant number of articles have focused on soybean and horseradish peroxidases (SBP and HRP, respectively), which have been among the top candidates that are being explored in this area”. How do these mentioned peroxidases help in degradation? Explain this term in the text.
Response: As requested, a new paragraph has been added which explains and shows the mechanism by which these peroxidases breakdown organic pollutants.
These peroxidases all share a common reaction cycle in which the iron (Fe3+) in the heme of the resting state reacts with hydrogen peroxide (H2O2) to form the oxo-Fe4+ cation radical form of the enzyme called Compound I. This can readily react with an organic substrate molecule to form an organic radical (Rl) and the oox-Fe4+ form of the enzyme called Compound II. The compound II form of the peroxidase can itself react with another organic substrate molecule to return to the resting form and create another organic radical (Rl), as shown below:
Peroxidase (resting state: Fe3+) + H2O2 à Peroxidase (Compound I: oxo-Fe4+l) + H2O
Peroxidase (Compound I: oxo-Fe4+l) + RH à Peroxidase (Compound II: oxo-Fe4+) + Rl
Peroxidase (Compound II: oxo-Fe4+) + RH à Peroxidase (resting state:Fe3+) + Rl + H2O
The organic radicals (Rl) then spontaneously react/breakdown to form smaller organic compounds [17].
Page 2: “Redox mediating species have also been used to improve the catalytic potential of some peroxidases for degrading some EPs”. How redox mediating species improve the potential of some peroxidases for degrading some EPs? Explain this term in the text. Also categorized the peroxidases and Eps regarding this context.
Response: We have explained the mode of action of redox mediators in the text, as follows:
Redox mediators, which are small, diffusible, redox-active organic compounds that act as “go-between agents” in peroxidase-catalyzed reactions, have also been shown to improve the catalytic potential of peroxidases for degrading some EPs [18,23]. Due to their small size and redox potential, these redox mediators preferentially react with the peroxidases and are converted into high reactive radical, which in turn can react and degrade different organic compounds or recalcitrant EPs.
The effect of redox mediator HOBT on EP degradation by specific DyPs are summarized in Supplementary Table 1S, and discussed in detail in section 4 (Results).
Page 2: “The continuous advancement of recombinant DNA technology has improved the potential for the successful and cheap production of active recombinant bacterial DyPs in Escherichia coli systems, unlike their eukaryotic counterparts, such as SBP and HRP, which are very difficult to produce in active forms using bacterial expression systems”. How these calculations have been performed explain thoroughly? Try to write a full form of abbreviation at the first sight.
Response: As suggested, we have included 3 recent references that show and discuss the efficient expression of recombinant bacterial peroxidases in E. coli.
Colpa DI, Fraaije MW. High overexpression of dye decolorizing peroxidase TfuDyP leads to the incorporation of heme precursor protoporphyrin IX. Journal of Molecular Catalysis B: Enzymatic. 2016 Dec 1;134:372-7.
Lambertz C, Ece S, Fischer R, Commandeur U. Progress and obstacles in the production and application of recombinant lignin-degrading peroxidases. Bioengineered. 2016 Apr 8;7(3):145-54.
Lončar N, Drašković N, Božić N, Romero E, Simić S, Opsenica I, Vujčić Z, Fraaije MW. Expression and characterization of a dye-decolorizing peroxidase from pseudomonas fluorescens Pf0-1. Catalysts. 2019 May;9(5):463.
Page 3: “After subjecting the EPs mixture to treatment with the seven rDyPs, samples were filtered using 0.45 μM 94 filters, prior to injecting them into the LCMS system”. Explain about the LCMS also write a full form of abbreviation at the first sight.
Response: As requested, the LCMS section in the Materials and Methods has been extensively rewritten and explained in details. All abbreviations used throughout the manuscript have been explained the first time they are mentioned. Thank you.
Page 4: “A 30-min elution run (in the MRM method) was developed where (i) 95% A and 5% B gradient was set for 5 min. (ii) a 5-90 % gradient of B was then adjusted for 20 min (from 5 min (95 % A) to 25 min (10 % A)) and another gradient of B % from 90 % 102 to 5% set for 10 seconds. (iii) at 25.10 min, 95 % A and 5 % B was set as a gradient for 4.90 min”. What is the MRM methods explain thoroughly?
Response: We have now included a detailed discussion of the MRM method, including a supplementary figure (S1) that explains it clearly.
Page 4: “The Mass spectrometry operating parameters for capillary voltage, nebulizer pressure, temperature of the gas and gas flow rate were set to 4000 V, 45 psi, 8 L/min, and 325 °C, respectively as previously reported”. Please site the relevant reference.
Response: Thank you for pointing this out. The relevant reference has now been added.
More recent research about types of wastewater pollution, treatment methods, various bi-composites and pollution reduction technologies is suggested to be added to make the background and discussion more strong: Journal of Water Process Engineering, 2020; 33:101101. Journal of Molecular Liquids, 2020; 313:113494. Journal of Environmental Chemical Engineering, 2020 Sep 19:104502. Journal of Environmental Management, 2020; 278:111302. International Journal of Biological Macromolecules, 2020; 161:1272-1285
Response: Thank you for the excellent suggestion. We have included 3 of the suggested (recent) references.
Page 6: “An amino acid sequence alignment of the seven DyPs that were used in this study are presented in Figure 1”. Figure 1 is missing in this Manuscript.
Response: We have made sure that all of the figures are now part of the main manuscript. Apparently, the figures (when uploaded separately) were not attached to the manuscript during our initial submission. Apologies for this!
The authors are advised to write the conclusions in a comprehensive way and should contain key values, suitability of the applied method, the major findings and contributions.
Response: As requested, the conclusion has been completely revised to make it more comprehensive.
The manuscript does not contain any figure. Please add some figures or convert table data into figures.
Response: We have made sure that all of the figures are now part of the main manuscript. Apparently, the figures (when uploaded separately) were not attached to the manuscript during our initial submission. Apologies for this!
The authors are advised to revise references, including the latest references. Please see some suggestions in the comments for the ‘introduction’ section.
Response: The introduction has been revised and as suggested, latest (2020) references have been added.
Reviewer 2 Report
Dear Editor, The authors did an excellent job. The content of the manuscript sounds perfect. Moreover, the English language and style are fine but spell checks are required. Thanks,
Author Response
Response to reviewers
Re: Biomolecules-1158090
Review #2:
Dear Editor, The authors did an excellent job. The content of the manuscript sounds perfect. Moreover, the English language and style are fine but spell checks are required. Thanks,
Response: Thank you! We have proofread the manuscript again and have improved the English language.
Reviewer 3 Report
The present manuscript aimed to study the degradation of 2-Mercaptobenothiazole using dye decolorizing peroxidases 19
(DyPs). This topic is important and fitted the scope of the journal. However, this wok is not well prepared and many aspects are missing. For instance, the details of influencing factors, degradation kinetics, mechanisms, and products need to be considered. Furthermore, the degradation experiments with field water samples was nor conducted.
Author Response
Response to reviewers
Re: Biomolecules-1158090
Review #3:
The present manuscript aimed to study the degradation of 2-Mercaptobenothiazole using dye decolorizing peroxidases 19
(DyPs). This topic is important and fitted the scope of the journal. However, this wok is not well prepared and many aspects are missing. For instance, the details of influencing factors, degradation kinetics, mechanisms, and products need to be considered. Furthermore, the degradation experiments with field water samples was nor conducted.
Response: Unfortunately, during the initial submission of the manuscript, all of the intended figures got left out! We sincerely apologize for this. We have made sure that all of the figures are now part of the main manuscript. These figures show many of the details requested, such as “influencing factors” (effect of redox mediator), “mechanisms” and “product identification”. We are confident that with the full set of figures, all of the above concerns will be satisfied.
Unfortunately, we are not able to get “field water samples” for our study, as access to wastewater treatment plants is strictly controlled and restricted by the UAE government.
Reviewer 4 Report
I do not feel capable of reviewing the manuscript in the current format. My version of the manuscript does not contain any of the mentioned figures. Thus, I had to stop reading the text at the result and discussion section.
The tables, while at least existing, are an absolute mess. Table 1 is hard to read due to the extreme slimness of the columns. There is no apparent order of the emerging pollutants - they are ordered neither by category (which I would prefer) nor alphabetically. Some of the structures are tiny and hardly readible (e. g. Roxithromycin, page 21) or distorted (e. g. Biochanin A, page 22). Non-standard abbreviations like NSAID (and MRM in Materials and Methods; AOP and MBT in Table 5) are not given in full.
Table 2 shows a maximum of ++ in my version. I do not understand why the legend explains degradation rates up to ++++. It is also not possible to see the sixth column and I am not sure that I acutally see the entirety of the others. The emerging pollutants are - again - seemingly randomly ordered and classified into three differently coloured categories without explanation.
What does the grey box in Table 4 indicate?
Table S1 is completely unreadible for me. Columns 2, 4, and 6 are not visible at all.
No reason for the choice of emerging pollutants is given. The title already contains an error: 2-Mercaptobenzothiazole.
I would like to point out that the material and method section would also strongly profit from proper formatting and proofreading. 'Biodegradation of emerging pollutants by recombinant DyPs' belongs before the respective LCMS-analysis ('LCMS based EP degradation assay'). What you are explaining in l. 118 is, if I understand it correctly, an extracted ion chromatogram (EIC). Why not use the standard term? 'Area under the peaks' (AUP) is normally called 'area under the curve' (AUC). What is l. 130 'H2O2 was incrementally added at 20-min intervals' supposed to mean?
These serious issues need to be rectified and the figures added to the manuscript before any decision regarding the manuscript is possible.
Author Response
Response to reviewers
Re: Biomolecules-1158090
Reviewer #4:
I do not feel capable of reviewing the manuscript in the current format. My version of the manuscript does not contain any of the mentioned figures. Thus, I had to stop reading the text at the result and discussion section.
Response: Unfortunately, during the initial submission of the manuscript, all of the intended figures got left out! We sincerely apologize for this. We have made sure that all of the figures are now part of the main manuscript.
The tables, while at least existing, are an absolute mess. Table 1 is hard to read due to the extreme slimness of the columns. There is no apparent order of the emerging pollutants - they are ordered neither by category (which I would prefer) nor alphabetically. Some of the structures are tiny and hardly readible (e. g. Roxithromycin, page 21) or distorted (e. g. Biochanin A, page 22). Non-standard abbreviations like NSAID (and MRM in Materials and Methods; AOP and MBT in Table 5) are not given in full.
Response: Again, this was due to the formatting of the tables (and PDF preparation) by the MDPI portal. We have made sure that complete, legible, and full tables are now included in the manuscript. All of abbreviations are also explained when they are first mentioned. The emerging pollutants are also re-arranged in Table 1 (as per categories).
Table 2 shows a maximum of ++ in my version. I do not understand why the legend explains degradation rates up to ++++. It is also not possible to see the sixth column and I am not sure that I acutally see the entirety of the others. The emerging pollutants are - again - seemingly randomly ordered and classified into three differently coloured categories without explanation.
Response: Unfortunately, the tables got messed up during the PDF generation by the portal. We have now fixed this. The three shades in Table 2 are also explained now in the table legends, as follows:
What does the grey box in Table 4 indicate?
Response: Thank you for highlighting this. The shaded cell has been removed from Table 4.
Table S1 is completely unreadible for me. Columns 2, 4, and 6 are not visible at all.
Response: Unfortunately, the tables got messed up during the PDF generation by the portal. We have now fixed this.
No reason for the choice of emerging pollutants is given. The title already contains an error: 2-Mercaptobenzothiazole.
Response: The panel of emerging pollutants was chosen to represent a wide range of pollutants, belonging to different categories. The spelling of 2-Mercaptobenzothiazole has been correct. Thank you!
I would like to point out that the material and method section would also strongly profit from proper formatting and proofreading. 'Biodegradation of emerging pollutants by recombinant DyPs' belongs before the respective LCMS-analysis ('LCMS based EP degradation assay'). What you are explaining in l. 118 is, if I understand it correctly, an extracted ion chromatogram (EIC). Why not use the standard term? 'Area under the peaks' (AUP) is normally called 'area under the curve' (AUC). What is l. 130 'H2O2 was incrementally added at 20-min intervals' supposed to mean?
Response: As requested, we have re-arranged the Materials and Methods section as per the suggestion.
The MRM mode of running LCMS is different than the EIC mode. The MRM method/approach is now more clearly explained, with a new supplementary figure (S1). Additionally, all mentions of AUPs have been replaced with AUCs. We have also fixed the ambiguous statement with regard to the addition of H2O2. Thank you!
These serious issues need to be rectified and the figures added to the manuscript before any decision regarding the manuscript is possible.
Round 2
Reviewer 1 Report
The authors have responsed well to those proposed questions by reviewers and carefully revised the whole paper according to the suggestions of reviewers step by step. Therefore, the quality of this paper has elevated greatly, hence I agree to publish the current version of the paper without any further changes.
Author Response
Thank you!
Reviewer 3 Report
accept
Author Response
Thank you!
Reviewer 4 Report
The manuscript has been thoroughly edited and corrected and is now in a state, where it can be properly reviewed.
The results regarding the optimal pH correctly discuss the increased optimal pH for the CboDyP due to the glutamate in the active site. No discussion or reason is given for the decreased optimal pH for the PfDyP B2. A (short) respective discussion should be added.
Part 3. DyPs-mediated degradation of emerging pollutants gives a good and thorough discussion for the differences in degradation by the different DyPs. What is not addressed at all are similarities and differences in the EP structures that might lead to or correspond with these differences in degradation. In my opinion, this is the main oversight of the whole manuscript. A comparison and discussion regarding the EP structure would elucidate what common structural elements are an advantage or disadvantage for the degradation by DyPs in general and for certain DyPs specifically. This would also mean that the findings of the manuscript could be used to hypothesize on the degradation of different, not tested EP. Since the authors state that they specifically chose a broad range of EP, they certainly attempted to gain such a general overview of DyP degradation capabilities.
In l. 456, the authors state that MBT was completely degraded by SviDyP and CboDyP and refer to Figure 4A and B. While the degradation seems to be complete according to Fig. 4A, Fig. 4B shows up to 10% remaining MBT. This discrepancy should be explained in the manuscript.
In l. 461, SviDyP (not correctly in italics) is given in parenthesis without any explanation as to why.
In l. 481, the authors refer to 60% homology between their most successful DyPs. It would be nice for the reader to have a comparison for this value: Is this the highest homology between all evaluated DyPs? Which ones are more closely related, if any? If none are, what are the next most homologous DyPs and how homologous are they? It is not possible to easily obtain these values from the given multiple sequence alignment in Figure 1A.
Part 5. LC-MSMS analysis of MBT intermediates generated by SviDyP correctly states that it is important to study the intermediates of EP degradation as those can also be harmful. Nonetheless, no statement regarding the toxicity of the detected degradation products is made. This has to be added to the manuscript.
In l. 603, references 44-46 are given for the data from Table 5. In the table itself, different references are given and these seem to be incorrect. Please correct this carefully.
Table S1 would strongly profit from color coding according to the changes from the results without HOBT to give the reader an easy and quick overview of the effects of HOBT-addition (e.g., reddish background: less degradation with HOBT, greenish background: more degradation with HOBT).
Figure 1A is very small and hard to read at 100% size. Is there any way to make it bigger? In addition, I could not identify any ‘green on window default color’ characters even though these are described in the legend. In general, I would strongly prefer if the authors only changed either the background or the character color for simplicity reasons.
Figure 2, 3, 4A, 5, and 6 are missing the unit (%) on the y-axis. In Figure 2, the names of the DyPs are also not correctly in cursive.
Figure 6A is missing the A.
In addition, some minor grammatical and formatting errors remain:
- The second paragraph of page 2 (ll. 70 to 81) contains a number of extra broad spaces. This can be found again and again throughout the manuscript.
- In Material and Methods, some of the subchapters are separated by an extra empty line, some are not. The same goes for the paragraphs in the Result and Discussion section.
- In l. 230, spaces between numbers and units are missing.
- In l. 234: It should be either /min or min-1.
- In l. 274: ‘or’ should probably be ‘over’.
Author Response
Thank you for your excellent comments. Kindly find attached our response detailing the incorporation of the suggested revisions.
